# Assessing the Ameliorative Effect of Selenium *Cinnamomum verum*, *Origanum majorana*, and *Origanum vulgare* Nanoparticles in Diabetic Zebrafish (*Danio rerio*)

**DOI:** 10.3390/plants11070893

**Published:** 2022-03-28

**Authors:** Rosa Martha Pérez Gutiérrez, José Guadalupe Soto Contreras, Felipe Fernando Martínez Jerónimo, Mónica de la Luz Corea Téllez, Raúl Borja-Urby

**Affiliations:** 1Laboratorio de Investigación de Productos Naturales, Escuela Superior de Ingeniería Química e Industrias Extractivas, Instituto Politécnico Nacional (IPN), Mexico City 07708, Mexico; josecontreras112104@gmail.com; 2Laboratorio de Hidrobiología Experimental, Instituto Politécnico Nacional (IPN), Escuela Nacional de Ciencias Biológicas, Carpio y Plan de Ayala S/N, Casco de Santo Tomás, Mexico City 11340, Mexico; fjeroni@ipn.mx; 3Laboratorio de Investigación en Polímeros y Nanomateriales, Escuela Superior de Ingeniería Química e Industrias Extractivas, Instituto Politécnico Nacional (IPN), Edificio Z-5 Planta Baja Del Gustavo A. Madero, Mexico City 07730, Mexico; mcoreat@yahoo.com.mx; 4Laboratorio de Microscopía Electrónica de Transmisión, Centro de Nanociencias y Micro-Nanotecnologías (CNMN), Instituto Politécnico Nacional (IPN), Mexico City 07758, Mexico; rborjauq@ipn.mx

**Keywords:** selenium nanoparticles, green synthesis, polyherbal formulation, zebrafish, type 2 diabetes

## Abstract

*Cinnamomum verum*, *Origanum majorana*, and *Origanum vulgare* have been used in traditional medicine for a long time to treat diabetes because of their promising therapeutic effects. The combination of these plants (COO) was tested to improve their efficacy using selenium nanoparticles (Se-COO-NPs) and gum Arabic (GA) as stabilizers for sustained release. Phenolic compounds of plants were identified using liquid chromatography–mass spectrometry (LC–MS/MS). GA-Se-COO-NPs were characterized by spectroscopic and microscopic methods and evaluated in diabetic zebrafish. The ultraviolet spectrum was assessed to confirm the formation of plasmon resonance at 267 nm. The obtained particle size of selenium nanospheres was 65.76 nm. They were maintained in a stable form for 5 months at 4 °C. Transmission electron microscopy (TEM) images demonstrated the presence of individual spherical nanoparticles. Fourier transform infrared spectroscopy (FT-IR) showed the interaction between COO extract and selenium, exhibiting good entrapment efficiency (87%). The elemental analysis of COO extract and GA-COO-SeNPs confirmed that NPs were obtained. The zebrafish were exposed to a high glucose concentration for two weeks, and type 2 diabetes and oxidative stress responses were induced. In diabetic zebrafish, treatment with NPs showed antilipidemic and hypoglycemic effects, high survivability, and reduced levels of glucose, reactive oxygen species (ROS), and lipids in the blood. This group this had a higher survivorship rate than the diabetic control. The results demonstrated that GA-Se-COO-NPs have high antidiabetic potential, most likely because of the synergic effects of phenolic compounds and Se nanoparticles.

## 1. Introduction

Type 2 diabetes mellitus (T2DM) is a metabolic disorder of lipids, carbohydrates, and proteins caused by defective insulin action and insulin secretion [1]. Previous studies have reported the relationship between oxidative stress and insulin resistance in the development of T2DM. Oxidative stress is implicated in the production of reactive nitrogen species (RNS), and reactive oxygen species (ROS) are accompanied by several diseases such as β-cell dysfunction, insulin resistance, and impaired glucose tolerance, leading to diabetic complications [2]. Metabolic syndrome is a cluster of risk factors, including high blood glucose, dyslipidemia, hypertension, and obesity, leading to an increased risk of T2DM and cardiovascular diseases. The search for alternative therapeutics is important in ethnobotanical areas, including medicinal plants and traditional products, to prevent and treat diabetes and its associated complications [3,4,5,6]. 

In recent years, the low effectiveness of commercial drugs has increased the interest in elaborating new drugs that can significantly decrease carbohydrate-hydrolyzing enzymes while avoiding side effects and having high therapeutic efficacy [7]. In recent reports, nanotechnology has revolutionized a wide array of health fields, including gene delivery, tissue engineering, imaging, artificial implants, drug delivery, and diagnostics [8]. In the same way, the use of nanoparticles (NPs) has increased because they are more biocompatible and less toxic than traditional drugs. As a result, they can increase drug effects compared to larger-sized particles and can decrease toxicity [9].

Selenium is part of selenoenzymes and is considered a nutritional element with high antioxidant properties that protects cells from oxidative stress injury, making extremely important to human health [10,11]. However, using selenium salts as food supplements is not recommended due to their cytotoxicity; therefore, the use of selenium nanoparticles (SeNPs) are a strategy to reduce the risk of toxicity of this element [12]. Consequently, the interest in SeNPs has increased considerably as a carrier of bioactive compounds to develop environmentally friendly systems that are non-toxic [13].

Several studies have been conducted on medicinal plants to control diabetes and its complications. Among them is *Cinnamomum verum,* which improves glycemic reduction and decreases insulin resistance and lipid metabolism, enhancing common complications related to diabetes [14]. *Origanum majorana* regulates gene expression, improves lipid and carbohydrate metabolism and insulin resistance, and restores renal and hepatic tissue architectures [15]. *Origanum vulgare* inhibits glycosylation and α-glucosidase activity, promotes glucose uptake, and reduces oxidative stress. In addition, it increases the expression of GLUT2 and inhibits the expression of CPY2E1 in E47 and HepG2 cells [16]. In this study, the green synthesis of selenium, *C. verum, O. majorana,* and *O. vulgare* nanoparticles is performed with gum Arabic (GA-Se-COO-NPs). This was performed to potentiate the hypoglycemia effect of selenium nanoparticles. 

## 2. Results and Discussion

### 2.1. Phytochemicals

The obtained results of the LC-MS/MS analysis to identify phenolic acids are shown in Figure 1. They were identified through comparison with reference compounds and analysis by UV–Vis of the sample. The phytochemical composition of the leaves of the three studied plants is shown in Table 1 and Figure 1. The antidiabetic effects of these plants can be attributed to their content of phenolic compounds, which have been researched for their antidiabetic potential, including ferulic acid, which can markedly decrease blood glucose levels, restore alterations in insulin signaling, reduce inflammatory cytokine release, and inhibit protein tyrosine phosphatase1B (PTP1B) expression [17]. Cinnamic acid improves pancreatic β-cell functionality, stimulates insulin secretion, and increases reductions in glucose uptake of hepatic gluconeogenesis [18]. p-Coumaric acid and gallic acid reduce oxidative stress, enhance glucose tolerance, increase the levels of PPARγ mRNA and adiponectin, and decrease the lipid profile and the level of TNF-α expression [19].

Chlorogenic acid possesses anti-inflammatory, hypoglycemic, hypolipidemic, and antioxidant properties [20]. Syringic acid improves both parameters of hyperglycemia and hyperinsulinemia [21]. Vanillic acid decreases the concentrations of glucose, insulin resistance, free fatty acid, and triglyceride [22]. Supplementation of protocatechuic acid to diabetic mice suppressed glycation associated with diabetic complications [23]. Coumarin ameliorates dipeptidyl peptidase-IV (DPP-IV) and decreases oxidative stress and hyperglycemia [24]. Caffeic acid treatment in diabetic mice showed a protective effect on the kidneys and liver and demonstrated hypolipidemic and hypoglycemic properties [25]. Rosmarinic acid in cases of hyperglycemia reduces plasma glucose levels and insulin sensitivity [26]. Eugenols stimulate skeletal muscle glucose uptake via activation of the GLUT4-AMPK signaling pathway and enhance insulin sensitivity [27]. Cinnamaldehyde treatment in diabetic rats reduces glucose levels and lipid homeostasis [28]. Pyrogallol inhibits α-glucosidase and is considered a potential drug [29]. Carnosic acid improves kidney damage, increases urine creatinine, and attenuates diabetes-induced albuminuria in STZ-induced diabetic mice [30]. In vitro gentisic acid assays inhibit α-amylase and α-glucosidase enzymes [31]. Chicoric acid decreases endothelial dysfunction through activation of the AMPK signaling pathway in diabetes [32]. The administration of salvianolic acid increases phosphorylated acetyl CoA carboxylase (p-ACC) protein expression, glycogen synthase protein expression, and peroxisome-proliferator-activated receptor alpha (PPARα), while in skeletal muscle it increases glucose transporter 4 (GLUT4) and phosphorylated AMP-activated protein kinase (p-AMPK) protein expression [33]. This study confirmed the antidiabetic properties of the phytochemicals identified in the COO extract.

### 2.2. Synthesis GA-COO-SeNPs

The GA, selenious acid, and ascorbic acid were submitted to two hours of reaction at room temperature. In the beginning, the reaction mixture presented a light yellow color, which gradually changed to a red color, indicating that the reaction had taken place (Figure 1) [34]. The crude reaction showed an intense red color due to the high content of selenium nanoparticles. 

### 2.3. Particle Size, Zeta Potential, and PDI Determination

The average particle size, zeta potential, and polydispersity index (PDI) were determined in a Zetasizer Nano ZSP manufactured by Malvern Instruments. The results showed that the average particle size of SeNPs (98.24 nm) with GA stabilizer was smaller than SeNPs without GA after preparation (189.1 nm). This difference between diameters suggests that GA is located on the surfaces of SeNPs, preventing their aggregation [35]. The dispersion light scattering (DLS) assay in the colloidal solution of GA-SeNPs showed a particle size distribution range of 19.21 to 112.31 nm, with an average size of 45.0 nm (Figure 2A), a PDI of 1.4, and a zeta potential value of −22.9 mV (Figure 2A), while the synthesized GA-COOSeNPs exhibited a distribution range of 42.44 to 154.04 nm, with an average size of 98.24 nm (Figure 2B). The PDI was 1.6, confirming the reduced aggregation of selenium nanoparticles. The zeta potential of the synthesized GA-COV-SeNPs had a value of −26.1 mV (Figure 2B). The zeta potential results for both systems confirm that the synthesized nanoparticles had a narrow size distribution with high stability. Comparing these results with previous studies [36], the synthesized nanoparticles in the present study were more stable because they presented higher negative values of the zeta potential and a narrower size distribution. The differences were attributed to the plants that were used for the synthesis of SeNPs and the content of OH groups in COO, which participate in stabilizing the nanoparticles.

### 2.4. Morphology Characterization by Transmission Electron Microscopy (TEM)

The TEM images of SeNPs covered with COO and 0.5% gum are shown in Figure 3A,B. The synthesis produced polydisperse and spheroid nanoparticles that exhibited a bimodal distribution with a value range of 43.6 to 61.4 nm. As shown in Figure 3A,B, the nanoparticles appeared well-dispersed in the gum Arabic matrix, which confirmed the participation of GA in stabilizing and dispersing SeNPs. Thus, GA was confirmed as a good stabilizer in the present work. The discrepancy between the sizes found by DLS and TEM could be attributed to the fact DLS measures the hydrodynamic diameter, whereas TEM explores the metallic core [37]. The results demonstrated that the particles of GA-COO-SeNPs had a uniform spherical shape and were well-dispersed in the colloidal system.

### 2.5. UV–Visible Spectra

SeNPs formation was observed by the color change when the SeO_2_ reacted with the ascorbic acid and was monitored by UV–Vis. The participation of GA as a stabilizer at a concentration of 0.5% was evaluated, with the UV–Vis spectra of the colloidal dispersions being recorded, using GA as a blank. TEM images demonstrated that the presence of GA plays an important role in the formation and stability of both dispersion systems of SeNPs. The UV–Vis spectra of COO SeNPs and GA-COO-SeNPs are shown in Figure 4. The COO extract displayed wide absorption bands in the range of 200–450 nm. This band is not presented with pure precursors, while SeNPs dispersions appear as a characteristic absorption peak at 267 nm. This absorption phenomenon can be explained by the surface plasmon resonance of nanoparticles [38]. When the SeO_3_^2−^ concentration is increased, the absorption intensity also increases. Thus, the band at the 267 nm position shifts as the concentration increases. The UV absorption spectra demonstrated that COO extract was successfully absorbed on the surfaces of SeNPs. This finding agrees with other works related to SeNP synthesis [39,40].

### 2.6. Fourier Transform Infrared Spectroscopy (FTIR)

FTIR analysis was used to support the formation of selenium nanospheres. There are similarities between the spectra of GA-COO-SeNPs, GA, and COO. Figure 5 shows absorption peaks of COO extract at 3406, 1748, and 1238 cm^−1^, associated with the stretching vibrations of hydroxyl, carbonyl, and C-O-C, respectively. However, in the spectrum of GA-COO-SeNPs, the signals (3406, 1748, and 1238 cm^−1^) are slightly shifted, as is the peak that shows the stretching vibrations of the hydroxyl group, which moved from 3406 cm^−1^ to 3443 cm^−1^. In addition, the associated signal of the carbonyl group changed from 1748 cm^−1^ to 1732 cm^−1^. The C-O-C group changed from 1238 cm^−1^ to 1249 cm^−1^, and the peak at 1648 cm^−1^ was displaced to 1622 cm^−1^. Compared with the obtained peaks of the COO spectrum, these displacements confirm the interaction between Se and COO extract through Se-O bonds.

These findings indicated that the hydroxyl groups from COO molecules and the GA form a chain-like intermediate through electrostatic interactions with the precursor SeO_3_^2−^ anions at the beginning of the reaction. Ascorbic acid reduces the SeO_3_^2−^ anions to Se in situ. As the reaction proceeds, the Se atoms are added to the Se core. The OH groups from COO and GA play a decisive role in the nucleation and growth of the nanoparticles, forming a stable nanosystem and avoiding further segregation and precipitation of selenium.

### 2.7. Storage Stability of GA-COO-SeNPs 

The formation of SeNPs in the presence of GA showed a red-orange color attributed to the size, resulting in a much more stable nanoparticle dispersion, which was transparent and showed no precipitation of material for approximately five months (Figure 6). The SeNPs without GA lost transparency and precipitate after one week, demonstrating that the addition of GA improved the stability of the nanoparticles. The nanoparticles remained dispersed during the storage period. This was corroborated by utilizing DLS measurements. The average particle size of GA-COO-SeNPs during this period did not exceed 100 nm, demonstrating again that the GA was also absorbed on the surfaces of SeNPs, preventing their aggregation. The surfaces of Se particles have a strong attraction to the terminal hydroxyl groups of GA, contributing to their stabilization. This coincides with a similar mechanism demonstrated with previously studied gums, including the gum rhamnogalacturonan isolated from the bark of *Cochlospermum Gossypium* [41], gum tragacanth from *Astragalus gummifer* [42], gum ghatti from *Anogeissus latifolia* [43], gum kondagogu from *Cochlospermum Gossypium* [44], and gum olibanum obtained from *Boswellia serrata* [45].

The entrapment efficiency of COO in the nanoparticles was 87%. The highest entrapment indicated a relatively large affinity between COO and the nanoparticle matrix.

### 2.8. Release of COO from Nanoparticles at Different pH Levels

In vitro drug release of COO from nanoparticles was performed in PBS (pH 5.5, 7.4, and 9.0) at 37 °C. The assays displayed similar release profiles, characterized for an initial period of fast release followed by a slower release period. The initial period within the first five hours was caused by adsorption of COO or desorption of the surface-bound nanoparticles. The second period was relatively slow and close to 30 h, attributed to the COO diffusion across the channels and pores of the nanoparticles. The samples at the different pH levels presented an initial burst effect during the first 5 h because of the adsorption of COO on the surfaces of the nanoparticles. After the fast release, a constant release was maintained. At pH 5.5 the total release was reached within 25 h and with an efficiency of 81%, while at pH 7.4 and 9.0, these efficiency levels were 75% and 66%, respectively. This difference could be attributed to these nanoparticles showing maximum swelling in acid pH, accelerating the COO release (Figure 7). The results show that the pH of the solution significantly affects the efficiency of COO release, which was performed over a broad pH range from 5.5 to 9.0 (Figure 7). The highest release efficiencies were shown at pH values from 5.5 to 7.4 for the release of COO, demonstrating a pH resistance to acid values. 

### 2.9. Elemental Analysis of COO Extract and GA-COO-SeNPs

The elemental analysis results for GA-COO-SeNPs displayed absorption peak characteristics of selenium at a proportion of 28.12% and of carbon and oxygen at 45.67% and 24.31%, respectively (Figure 8). However, COO extract mainly presented carbon and oxygen elements at proportions of 59.38 and 36.27%, respectively. Furthermore, the analysis revealed that Se gum had a conjugation rate of 28.12%, showing that selenium nanoparticles were successfully synthesized with COO extract.

### 2.10. Effects of GA-COO-SeNPs on the Survival Rate of Zebrafish 

The effect of GA-COO-SeNPs on the survival rate of zebrafish was carried out to determine the possible toxicity of nanospheres at concentrations of 10−30 mg·L^−1^. The zebrafish treatment of GA-COO-SeNPs remained close to 100% survivorship (Figure 9A). The survival rate reduced slightly to 6% with the increment in concentration from 50 to 100 mg·L^−1^, with this mortality rate being not significant. In summary, the observed data demonstrated that concentrations less than 100 mg·L^−1^ were not toxic to zebrafish. 

### 2.11. Biological Activity of GA-COO-SeNPs in Zebrafish

After animals fasted for 24 h, blood samples were collected from the zebrafish caudal vein and their biochemistry parameters were measured. The findings indicated marked increases in blood glucose, total cholesterol (TC), and triglyceride (TG) concentrations and a reduction in antioxidant enzymes provoked by the exposure to glucose. The NPs of the polyherbal formulation of *C. verum, O. majorana,* and *O. vulgare* extracts stabilized with GA (GA-COO-SeNPs) had a hypoglycemic and antilipidemic effect and decreased oxidative stress in zebrafish previously induced to the diabetic conditions through transdermal exposure to glucose, as shown in Table 2. Serum glucose, total cholesterol (TC), and triglyceride (TG) levels of groups exposed to glucose were increased by around 3.08-, 2.85-, and 2.44-fold compared with the control group, respectively. In zebrafish exposed to COO extract and selenium nanospheres (GA-SeNPS, GA-COO-SeNPs) at a dose of 20 µg/L and to Metformin (20 mM), blood glucose levels were significantly reduced by about 67.0, 46.5, 72.8, and 64.33%, respectively compared with the glucose-induced diabetic zebrafish group. The concentrations of plasma TC and TG in the normal and glucose-induced diabetic zebrafish control group are also shown in Table 2. 

Hypercholesterolemia was significantly decreased in the COO, GA-SeNPS, and GA-COO-SeNPs groups at the 20 µg/L concentration compared with the diabetic groups. The reduction effect was remarkably higher in the GA-COO-SeNP group than in the diabetic control group. The regulation of these biochemical parameters in zebrafish demonstrated its similarity to those observed in mammals. Administration of GA-COO-SeNPs to diabetic fish decreased the levels of plasma glucose TC (65.3%) and TG (41.5%), similar to the findings observed in the control groups. The results demonstrated the antidiabetogenic and antidyslipidemic effects of these extracts in the SeNPs for fish-induced diabetic conditions. Previous studies have shown that selenium nanoparticles exhibited biological activity in streptozotocin-induced diabetic rodents [46] and other pharmacological models of diabetes [47]. The finding demonstrated that SeNPs could ameliorate the poor aqueous stability and solubility of the extracts to control diabetes by hauling bioactive extracts for slow release. The use of nanoparticles could be a potential treatment in the control of T2DM through the possible synergic effects of bioactive extracts containing phenolic components with SeNPs. Thus, GA-COO-SeNPs may be promising delivery nanocarriers and may have therapeutic potential for treating this disease. 

### 2.12. Effects of GA-COO-SeNPs in Antioxidant Enzymes of the Liver in Zebrafish

In this study, no dead or sick fish were observed during the exposure or treatment periods. The activities of CAT, SOD, and GPx enzymes in the liver of zebrafish exposed to glucose are shown in Figure 9B–D. These are used as indicators of oxidative stress. Diabetic groups after 7 and 14 days reduced significantly (*p* < 0.01) in terms of CAT, SOD, and GPx activities compared with the control group at 8 and 62.5%; 29 and 53.2%; and 35 and 47.8%, respectively. Enzyme inactivation can be caused by excessive generation of ROS that antioxidant defense cannot eliminate. This process begins with the transformation by SOD of oxygen-free radicals to H_2_O_2_ molecules, then GPx and CAT degrade these peroxides into H_2_O and oxygen molecules, which are indispensable inadequately maintaining the intracellular redox balance. However, during exposure (7 and 14 days) to COO, GA-SeNPS, and GA-COO-SeNPs, ROS production was neutralized, inhibiting oxidative stress. In contrast, in the control individuals, an increment in activities of antioxidant enzymes after 14 days of treatment was observed (Figure 9B–D). The treatment with GA-COO-SeNPs at a dose of 20 µg/mL produced the best results, significantly increasing (*p* < 0.05) the levels of SOD (2.22-fold), CAT (3-fold), and GPx (2.04) in the liver. The activity of antioxidant enzymes in the metformin-treated group was much lower than that of the COO-GA-SeNPs treated group. These data indicated that treatment with GA-COO-SeNPs could improve the antioxidant capability of zebrafish. The findings demonstrated that the protective effect of COO-GA-SeNPs can be explained by the upregulation of SOD, CAT, and GPx activities in the liver, providing protection from oxidative injury to the cellular constituents. Oxidative stress is associated with complications in diabetic individuals [48].

## 3. Materials and Methods

### 3.1. Generals

All reagents used in this study were purchased at analytical grade from Sigma Aldrich (St. Louis, MO, USA).

### 3.2. Plant Materials 

The fresh leaves of *C. verum, O. majorana*, and *O. vulgare* were obtained from Central de Abastos in CDMX, México, and authenticated in the Department of Botany by Graciela Calderón; voucher specimens were deposited in the Herbarium of National School of Biological Sciences—IPN for further reference (2020-1078, 2020-1082, 2020-1080, respectively). The plants were dried in the shade at room temperature, and leaves were ground in an electric grinder to pass through a 50-mesh sieve. A polyherbal formulation containing the combination of *C. verum, O. majorana,* and *O. vulgare* in proportions of 1:1:1 *v/v* were prepared.

### 3.3. Ultrasound-Assisted Extraction (UAE) of Polyphenolic Compounds (COO)

Ten grams of polyherbal formulation powder was placed in a capped tube and subjected to UAE using an ultrasonic cleaning bath (KH5200 DB type, Kunshan ultrasonic instrument Co., Ltd., Kunshan, China). The process was carried out using a frequency of 40 kHz and at 200 W with the following optimal extraction conditions: ethanol 66.03%, 28.87 min, and 21.51 mL/g for maximal flavonoids extraction [49]. After we collected the supernatant using UAE, the samples were centrifuged at 5000 rpm for 10 min and UV–Vis analyses were performed at 360 nm. 

### 3.4. Total Polyphenolic Content

The total polyphenolic content was evaluated using the Folin–Ciocalteu assay, according to the European Pharmacopoeia [39]. Two milliliters of COO extract were diluted 25 times with distilled water and mixed with Folin–Ciocalteu reagent (1.0 mL), which was diluted to 25.0 mL with a solution of sodium carbonate (Na_2_CO_3_; 290 g/L). Absorbance was measured after 30 min at 760 nm, and the results were shown as mg of gallic acid equivalent (GAE)/g dried plant material obtained from a calibration curve of gallic acid (R2 = 0.996).

### 3.5. Identification of Phytochemicals 

LC-MS/MS was used to analyze the extracts of the plants. *C. verum, O. majorana*, and *O. vulgare* were carried out on Agilent series 1290 UHPLC system consisting of a quaternary pump, autosampler, and thermostatic column compartment (Agilent Technologies, Santa Clara, CA, USA). Phenolic profiles were achieved using a C18 analytical column (2.1 mm × 100 mm, 3.5 μm). Elution was carried out with water–formic acid (0.1%) (A) and water–acetonitrile (B) as the mobile phases, starting with a gradient of B 10−18% (0–4 min), 18−20% B (4−9 min), and 20−20% B (9−10 min). Chromatograms were registered at 280 nm and were established by their retention time compared with those of the reference standards. MS spectra were assessed using an Agilent 6460 triple quadrupole mass spectrometer equipped with an electrospray ionization (ESI) source (Agilent Corporation, Santa Clara, CA, USA). The full scan mass covered the range from 100 up to 1500 *m*/*z*.

### 3.6. Synthesis of Selenium-Nanoparticle-Embedded Gum Arabic Microspheres

From selenous acid (H_2_SeO_3_), COO-SeNPs were prepared using a chemical reduction [50]. Briefly, COO was dissolved to a 4 mg/mL final concentration in Milli-Q water. Next, 20 mL of COO solution was combined with selenous acid solution (4 mM) at a ratio of 1:2 with gum Arabic (GA; 0.5%) under magnetic stirring. Then, ascorbic acid at a concentration of 8 mM in 40 mL of Milli-Q water was added and stirring was maintained in the dark for 24 h at 25 °C. The formed GA-COO-SeNPs were separated by centrifugation at 15,000 rpm for 15 min and redissolved in a suitable volume of Milli-Q water. This process was repeated three times to eliminate any excess chemicals remaining in the solution

### 3.7. Characterization of GA-COO-SeNPs

Diluted nanoparticles solutions were measured using transmission electron microscopy (TEM) (HT7700, Hitachi High-Tech, Tokyo, Japan) at an accelerating voltage of 80 kV. Briefly, TEM nanoparticles were prepared by directly drying a drop of the pre-treated GA-COO-SeNPs solution onto a carbon-film-coated copper grid and dried in air for 5 min. Dynamic light scattering (DLS, Zetasizer Nano ZSP, Malvern Instruments Ltd., Malvern, UK) was used to monitor particle size changes during storage. Between 1.0–1.5 mL of each sample was measured in a polystyrene cuvette at a fixed angle of 173° at 25 °C. Laser Doppler velocimetry (LDV, Zetasizer Nano ZS, Malvern, UK) was applied to obtain zeta potential values to check the bonding mechanism. The non-invasive backscattering (NIBS) technique was used to evaluate the particle size distribution. The results are reported as the average values of triplicate measurements.

The UV–Vis absorption spectra of nanoparticle solutions were monitored with a UV-250 Shimadzu spectrophotometer (Tokyo, Japan) at wavelengths of 250–700 nm.

FTIR (TENSOR27, Bruker) was used to measure the chemical group of nanoparticles in the range of 400–4000 cm^−1^.

Energy-dispersive X-ray spectroscopy (EDS) was performed to detect the elemental composition of GA-COO-SeNPs.

### 3.8. Encapsulation Efficiency (EE%) 

Briefly, COO-loaded nanoparticles were centrifuged at 15,000 rpm for 30 min. The absorbance of the supernatant was measured at 280 nm. The absorbance was transformed into standard calibration plots of the quantity of COO utilization [51]. The EE% was measured using the following equation (1): EE (%) = (Total drug-free drug)/(Total drug) ×100(1)

### 3.9. COO Release In Vitro

GA-COO-SeNPs were added to phosphate-buffered saline (PBS, pH 7.4), then placed in a dialysis bag (3500Da, Thermo Scientific, Rockford, IL, USA), immersed in 10 mL of PBS in a glass vial (50 mL), and incubated at 37 °C for one h. Next, 1 mL of the sample at time intervals of 5, 10, 20, and 30 h was used to measure COO release, then replenished with the same amount of fresh PBS. The same analysis was performed at pH 5.5 and 9.0. The absorbance of the aliquots was measured at 280 nm. The concentration of COO was evaluated using a calibration curve

### 3.10. Maintenance of Zebrafish (Danio rerio)

Zebrafish were placed in tanks at an adequate density, avoiding overcrowding (20 adults/5 L), in dechlorinated tap water (pH 6.89), at a temperature of 28 ± 1 °C and photoperiod of 14 h (light)/10 h (dark). Adequate oxygen levels were maintained using air pumping and by changing the culture medium every other day. Methylene Blue was added to the tank to avoid the growth of fungus, and the animals were fed with micro pellets (balanced food) [52]. All zebrafish procedures were approved by the Ethics Committee of the Escuela Nacional de Ciencias Biológicas-IPN, complying with international criteria.

### 3.11. The Survival Rate of Zebrafish

The effect of GA-COO-SeNPs on the survival rate of zebrafish was assessed to determine the toxicity of nanoballs at concentrations of 10−100 mg·L^−1^

### 3.12. Assessment of the Antidiabetic Effect of Nanoparticles

Zebrafish were induced to type 2 diabetic condition by maintaining organisms in 5 L tanks containing dechlorinated tap water with glucose 110 mM added continuously for two weeks (20 adult zebrafish per tank, whereas control, non-diabetic fish were maintained in dechlorinated tap water at the same density). Then, to evaluate the permanence of the diabetic condition, glucose-exposed individuals were placed in clean, fresh water for one week. Finally, some of these fish were sacrificed by hypothermia on ice and their biochemical parameters were determined. The findings showed that blood glucose level was increased three-fold in comparison to the normal control group. The short induction time of this method and the persistence of hyperglycemic conditions are both important advantages of this procedure [53].

### 3.13. Biochemical Analysis

After the exposure time, blood samples were collected and centrifuged at 3000× *g* for 15 min to obtain the supernatant for the biochemical assay for fish in all treatments. The glucose level was evaluated using an Accu-Chek Glucometer (Accu-Chek^®^ Active, Roche Diagnostics GmbH, Mannheim, Germany) and expressed as mg/dL. In individuals with blood glucose levels over 95 mg/dL (5.27 mM/L), diabetes was confirmed. The concentrations of high-density lipoprotein (HDL), total cholesterol, and triglycerides in the serum samples of the fish were measured using the kit assay (total cholesterol and triglycerides measured using a Home Test Meter Kit Monitor, Solana Health Inc., Del Mar, CA, USA). Liver samples were individually taken from three adult zebrafish after 14 days of exposure to GA-COO-SeNPs to measure oxidative stress biomarkers in the serum; the antioxidant enzymes superoxide dismutase (SOD), catalase (CAT), and glutathione peroxidase (GPx) were evaluated using the kit assay from Abcam (México) according to the manufacturer’s instructions.

### 3.14. Statistical Analysis 

The statistical analysis of the results of glucose-induced zebrafish diabetes are presented as mean DS values, implicating a one-way analysis of variance (ANOVA). Tukey’s multiple comparison test was employed to compare diabetic control, normal control, and treated groups with NPs to analyze the data (GraphPad Prism version 7, GraphPad Software, San Diego, CA, USA). Statistical significance vs. diabetic group is indicated by * *p* < 0.001, ** *p* < 0.01, and *** *p* < 0.05, whereas # indicates comparison to the control group.

## 4. Conclusions

In this study, SeNPs were synthesized with an easy sol–gel method by reducing selenous acid to elemental selenium and stabilizing it with gum Arabic. The synthesis of nanoparticles was detected by a visual change in color of the extract from light yellow to orange-red. The COO extract from *Cinnamomum verum, Origanum majorana*, and *Origanum vulgare* was rich in phenolic and flavonoid compounds, which was suitable for the biosynthesis of nanoparticles. The synthesized GA-COO-SeNPs were highly stable and had a spherical shape and nanometer size. The zeta potential of the nanoparticles was negative. Administration of GA-COO-SeNPs significantly ameliorated blood sugar, antioxidant enzymes activity, and lipid levels in glucose-induced diabetic zebrafish, indicating its significant antidiabetic activity. In addition, the *C. verum, O. majorana,* and *O. vulgare* combination had a higher effect on the antidiabetic parameters than the single extract of each plant. Additionally, values were higher in GA-COO-SeNPs, suggesting that the nanoformulation of the herbal formulation in Se, added with GA, increased the therapeutic activity and reduced the levels of glucose, cholesterol, and triglycerides while improving the antioxidant activity. The extracts of these plants, supplied as nanoparticles added to gum Arabic, could be a promising natural therapy to control the injuries related to diabetes mellitus disease.

## Data Availability

Not applicable.

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
