# Peer review of "Assessing the Ameliorative Effect of Selenium Cinnamomum verum, Origanum majorana, and Origanum vulgare Nanoparticles in Diabetic Zebrafish (Danio rerio)"

_plants, 2022, doi:10.3390/plants11070893_

Round 1
Reviewer 1 Report
The manuscript by Gutiérrez et al utilize green synthesis approach for synthesis of Se nanoparticles and explored their biological potency as anti-diabetic materials. However, the study is poorly designed and presented. The author need to improvise the contents drastically.
- Author should add additional information about material characterization, about stability and reusuability of the material. The author should also perform metabolite analysis in the nanoparticle and compare the peak pattern in crude extract and in nanoparticles.
- The quality of image should e high quality.
- Why diabetic only, why not check antimicrobial or cytotoxic and chemical degradation assays?
Author Response
d
Comments and Suggestions for Authors
The manuscript by Gutiérrez et al utilize green synthesis approach for synthesis of Se nanoparticles and explored their biological potency as anti-diabetic materials.
However, the study is poorly designed and presented. The author needs to improvise the contents drastically.
The entire manuscript was improved as suggested the reviewer 2:
Also, ensure a finding is discussed in relation to other findings ie. indicating how these support each other and thus the use thereof. Expand on discussion relative to literature
- Author should add additional information about material characterization, about stability and reusability of the material.
About the stability of the nanoparticles, it is very well determined by the UV-vis, Zeta sizer, and storage studies which are explained more fully.
Reusability of the nanoparticles is used in practical engineering applications
and enzymes mainly, thus the effective degradation cycles of nanoparticles
are very important in the degradation of pollutants and in situ environmental remediation. We consider that since our nanoparticles are subjected to metabolic processes, it is not necessary to determine
- The author should also perform metabolite analysis in the nanoparticle and compare the peak
pattern in crude extract and in nanoparticles.
Metabolite analysis of C, H, and Se were added
- The quality of image should e high quality.
The Figs were changed
- Why diabetic only, why not check antimicrobial or cytotoxic and chemical degradation assays?
Since one of the most important properties of the 3 plants is the antidiabetic potential, that is why it was decided to study this disease. Although they are also reported for their cytotoxic and antimicrobial effects, it is not our area of expertise.

Reviewer 2 Report
Dear Editor,
First of all I would like to thank to MDPI for inviting me to review this paper.
I carefully read the submission title 'Antidiabetic potential of selenium nanoparticles embedded gum Arabic decorated with the polyherbal formulation of Cinnamomum verum, Origanum majorana, and Origanum vulgare in glucose-induced diabetic zebrafish (Danio rerio)'.
In fact more recently there has been an increasing interest to medicinal plants ant its products.
My first impression that the paper contain new information and title of the manuscript cover its content. The summary is appropriate and the aim of the work clearly established. The methods are used are adequate and used sophisticated techniques and equipment's. I found the results very reliable. Discussion and conclusions are well documented and scientifically coherent.
However, I have some corrections and additions on it before acceptance.
TITLE: Should be shortened and in its current form it is long.
ABSTRACT: The first sentence must be rewritten as
Cinnamomum verum, Origanum majorana, and Origanum vulgare are used in the treatment of diabetes When??? traditionally??? please be more specify.
The abbreviation COV is not gave a sense should be changed
Do not use abbreviations when first time used such as SeNPs, FTIR, TEM etc
Please correct as Results showed that
INTRODUCTION:
Line 56.
Selenium is part of selenoenzymes and is considered a nutritional element with elevated antioxidant properties by protecting cells from oxidative-stress injury which are extremely important to human health [7]. Give one more reference I suggest below one
Dong CY. Effects of biological metabolism of Metasequoia glyptostroboides on nutrient element content and enzyme activity in seedling soil. Turk. J. Agric. For. 2021, 45, 642-650.
Line 63. Numerous studies have reported that medicinal plants have effects hy- poglycemic and hypolipidemic on diabetic mice that exhibited significant antidiabetic and ant obesity activities [3].??? Please gave more references......
Figure 2 is not clear and resolution should be increased
Who authenticated the plant samples????
Conclusion relatively short.
Author Response
Dear Editor,
First of all I would like to thank to MDPI for inviting me to review this paper.
I carefully read the submission title 'Antidiabetic potential of selenium nanoparticles embedded gum Arabic decorated with the polyherbal formulation of Cinnamomum verum, Origanum majorana, and Origanum vulgare in glucose-induced diabetic zebrafish (Danio rerio)'.
In fact more recently there has been an increasing interest to medicinal plants ant its products.
My first impression that the paper contain new information and title of the manuscript cover its content. The summary is appropriate and the aim of the work clearly established. The methods are used are adequate and used sophisticated techniques and equipment's. I found the results very reliable. Discussion and conclusions are well documented and scientifically coherent.
However, I have some corrections and additions on it before acceptance.
TITLE: Should be shortened and in its current form it is long.
The title of the article was reduced: Selenium Cinnamomum verum, Origanum majorana, and Origanum vulgare nanoparticles (Se-CONPs) and their ameliorative effects in diabetic zebrafish
ABSTRACT: The first sentence must be rewritten as
Cinnamomum verum, Origanum majorana, and Origanum vulgare are used in the treatment of diabetes When??? traditionally??? please be more specify.
The use of plants was related to traditional medicine
The abbreviation COV is not gave a sense should be changed
The abbreviation COV was changed to the initials of the scientific names of each plant (CO)
Do not use abbreviations when first time used such as SeNPs, FTIR, TEM etc
In the abstract, each abbreviation was given its meaning
Please correct as Results showed that
The sentence was changed
INTRODUCTION:
Line 56.
Selenium is part of selenoenzymes and is considered a nutritional element with elevated antioxidant properties by protecting cells from oxidative-stress injury which are extremely important to human health [7]. Give one more reference I suggest below one
Dong CY. Effects of biological metabolism of Metasequoia glyptostroboides on nutrient element content and enzyme activity in seedling soil. Turk. J. Agric. For. 2021, 45, 642-650.
The reference was added to the text
Line 63. Numerous studies have reported that medicinal plants have effects hy- poglycemic and hypolipidemic on diabetic mice that exhibited significant antidiabetic and ant obesity activities [3].???
Please gave more references......
The other three references were added to the text
Figure 2 is not clear and resolution should be increased
Fig was modified
Who authenticated the plant samples????
The name of the Biol. Graciela Calderón was added to text
Conclusion relatively short.
The conclusion made more longer
English language requires improvement.
The English language was revised
Title: encapsulated rather than decorated
The title was changed
Abstract: minimize the methodology part and increase findings backed up with values.
The methodology was minimized
Plants: voucher numbers are required. What were the ratios of each plant to produce the polyherbal formula?
The requested data were added in the text
Results and Discussion:
some sections are too short. It is recommended that these are grouped together with other sections.
Also, ensure a finding is discussed in relation to other findings ie. indicating how these support each other and thus the use thereof. Expand on discussion relative to literature
Discussion was expanded
If there has been a change in the authorship during revisions of your paper, please download the "Authorship Change Form" to provide details of the change, then please upload it together with your resubmission.
Changes were downloaded to the "Authorship Change Form"

Round 2
Reviewer 1 Report
The revised version is highly improvised. However, I would like to recommend the author to perform some additional ROS quantification assays either by use of fluorescence and chemiluminescence probes, and electron spin resonance (ESR/EPR). I would be highly appreciate if author can discriminate and quantify each type of ROS.
Author Response
Comments and Suggestions for Authors
The revised version is highly improvised. However, I would like to recommend the author to perform some additional ROS quantification assays either by use of fluorescence and chemiluminescence probes, and electron spin resonance (ESR/EPR). I would be highly appreciated if author can discriminate and quantify each type of ROS.
Reviewer proposed determination of ROS by a fluorescence method can serve as an effective tool to study dynamic redox chemistry in living systems and will provide inspiration to facilitate diagnostics and therapies for the treatment of diseases. This fluorescence method determines reactive oxygen species (ROS) production as a by-product of mitochondrial activity in consequence NPs must be probed in cells.
The chemiluminescence method is luminol-based has been used for western
blotting rather than measuring cell-derived ROS. Luminol detect low concentrations of ROS in nonimmune cells activity in consequence NPs must be probed in cells to measure
ROS released by cells into media and use cell lysates to investigate intracellular signaling pathways.
The electron spin resonance (ESR/EPR)
To prove that indeed oxygen radicals generated during activation of B[a]P in rat
lung microsomes, liver microsomes, oxidative DNA damage, and B[a]P-DNA Adducts marked with [Υ- 32 P]-ATP. In addition to performing these studies, a spectrometer equipped with an ER 4119HS high sensitivity cavity and 12 kW power supply operating at X band frequencies will be used and the quantitation of the spectra will be performed by peak height measurements
using the WIN-EPR spectrum manipulation program. Working with radioactive material requires special equipment.
Dear Editor:
The English of the article was reviewed by a native of the USA
Respectfully we disagree with the reviewer's comments. The study we performed could be criticized because it is not written in perfect English, but in no way, it is improvised at all; this comment of the reviewer could arise because of an erroneous understanding of the objective of our study. Our main objective was to develop Se nanoparticles with an herbal extract formulation and to characterize these nanoparticles to determine their antidiabetic effect as a way to propose an alternative, no conventional treatments against a high-impact public health problem, as is diabetes. In this respect, the measure of oxidative stress was only with the aim to obtain information of one of the consequences of this disease and to observe the modification of this response linked to the use of the alternative recovery treatment.
The methods suggested for the reviewer to quantify ROS are of course adequate for an in-depth analysis of these molecules but are out of the scope of our study; instead, we think that was more relevant to confirm the recovery of the glucose levels in diabetic-induced fish. The measure of the antioxidant activity and the concentration of triglycerides as we did were additional determinations to confirm the anti-diabetic effect of the novelty treatment applied to diabetic fish, but the application of additional, more complex methods is out of the scope of our research but could be adequate for another study with emphasis in oxidate stress at the cellular and subcellular level.
